# Enhanced Sensitivity to ALDH1A3-Dependent Ferroptosis in TMZ-Resistant Glioblastoma Cells

**DOI:** 10.3390/cells12212522

**Published:** 2023-10-25

**Authors:** Yang Wu, Sophie Franzmeier, Friederike Liesche-Starnecker, Jürgen Schlegel

**Affiliations:** 1Department of Neuropathology, Institute of Pathology, School of Medicine, Technical University Munich, 81675 Munich, Germany; ge46muc@tum.de (Y.W.);; 2Department of Neuropathology, Institute for Animal Pathology, Ludwig-Maximilians-University Munich, 80539 Munich, Germany; 3Pathology, Medical Faculty, University of Augsburg, 81656 Augsburg, Germany

**Keywords:** therapy resistance, therapeutic interventions, glioblastoma, ferroptosis, ALDH1

## Abstract

Temozolomide (TMZ) is standard treatment for glioblastoma (GBM); nonetheless, resistance and tumor recurrence are still major problems. In addition to its association with recurrent GBM and TMZ resistance, ALDH1A3 has a role in autophagy-dependent ferroptosis activation. In this study, we treated TMZ-resistant LN229 human GBM cells with the ferroptosis inducer RSL3. Remarkably, TMZ-resistant LN229 clones were also resistant to ferroptosis induction, although lipid peroxidation was induced by RSL3. By using Western blotting, we were able to determine that ALDH1A3 was down-regulated in TMZ-resistant LN229 cells. Most intriguingly, the cell viability results showed that only those clones that up-regulated ALDH1A3 after TMZ withdrawal became re-sensitized to ferroptosis induction. The recovery of ALDH1A3 expression appeared to be regulated by EGFR-dependent PI3K pathway activation since Akt was activated only in ALDH1A3 high clones. Blocking the EGFR signaling pathway with the EGFR inhibitor AG1498 decreased the expression of ALDH1A3. These findings shed light on the potential application of RSL3 in the treatment of glioblastoma relapse.

## 1. Introduction

Glioblastoma (GBM) is characterized by its rapid and infiltrative growth and, consequently, its high rates of recurrence. Despite significant advancements in surgery, radiation therapy, and chemotherapy, the median survival rate remains discouragingly low, with an average survival of 12 to 15 months after diagnosis and a five-year survival rate of 7.2% [1]. The diffusely infiltrative nature of GBM makes complete removal by surgery nearly impossible, and the most disappointing aspect of GBM is its resistance to conventional therapies. Thus, the future of cutting-edge glioblastoma treatment approaches lies in precision medicine and customized medicines.

Temozolomide (TMZ) is part of the standard treatment regimen for GBM. It is usually given concomitantly with radiation therapy, followed by maintenance therapy with TMZ alone [2]. The methylation of the O-6-methylguanine-DNA methyltransferase (MGMT) promoter is a strong predictor of the therapeutic efficiency of TMZ treatment [3]. However, for reasons unrelated to their MGMT statuses, patients may undergo tumor relapse and develop resistance to TMZ [4,5]. TMZ generates reactive oxygen species (ROS) as byproducts of its metabolism and DNA-damaging effects, which contribute to oxidative stress in GBM cells [6]. Ferroptosis is an iron-dependent form of regulated cell death characterized by the accumulation of lipid peroxidation [7]. By inducing ROS generation, TMZ may potentially sensitize cancer cells to ferroptosis.

Aldehyde dehydrogenase (ALDH) 1A3 is a member of the ALDH enzyme family, and it plays a vital role in cellular detoxification and the oxidative stress response [8]. ALDH1 is primarily involved in the oxidation of retinaldehyde to retinoic acid, a crucial metabolite involved in various cellular processes, including differentiation and proliferation [9]. ALDH1A3 has been linked in recent studies to resistance to TMZ therapy [10] and to higher levels in recurrent GBM [11]. Moreover, the high expression of ALDH1A3 has been found to contribute to the maintenance of glioma stem-like cells (GSCs), a subpopulation of GBM cells with self-renewal capacities and enhanced resistance to therapies [12]. ALDH1A3 is regulated in many ways. In addition to direct or indirect transcriptional regulation mediated by different signaling pathways [13,14,15], the expression of ALDH1A3 is intricately regulated by histone epigenetic modifications at its promoter region [16]. Furthermore, post-transcriptional processes including autophagy or regulation by microRNA further contribute to its regulation [10,17].

The epidermal growth factor receptor (EGFR) plays an important role in the development and progression of GBM. EGFR activation triggers a cascade of signaling events, including the PI3K/AKT pathways, regulating cell growth, survival, and migration. The aberrant activation of AKT due to EGFR dysregulation contributes to the uncontrolled growth and aggressiveness of glioblastoma cells [18].

Emerging evidence has implicated the role of ALDH1A3 in conferring resistance to TMZ, accompanied by its influence on the autophagic process. Recent studies have placed significant emphasis on the association between ALDH1A3 and reactive oxygen species (ROS) metabolism, as well as its role in sensitizing glioblastoma cells to ferroptosis. The induction of ferroptosis appears to be a promising novel therapeutic strategy in the context of glioblastoma therapy.

Our previous results showed that TMZ induces ROS accumulation [8] and ALDH1a3 sensitizes GBM to ferroptosis [19]. Here, we demonstrated that TMZ-resistant (TMZ-res) LN229 clones were resistant to ferroptosis, although lipid peroxidation was induced by the ferroptosis inducer RSL3. We further discovered that ALDH1A3 was downregulated in TMZ-resistant LN229 clones in an EGFR-dependent PI3K-pathway-mediated manner. After the discontinuation of TMZ, cells up-regulated ALDH1A3 and became re-sensitized to ferroptosis induction. This might provide a glimpse into the potential future application of ferroptosis in glioblastoma treatment.

## 2. Materials and Methods

### 2.1. Cell Culture, Reagents, and Antibodies

The LN229 glioblastoma cell line (PTEN wild-type, p53 mutation, MGMT activity-deficient) (10.1111/jnc.14262) was obtained from American Type Culture Collection (ATCC, Manassas, VA, USA) and cultured in Dulbecco’s Modified Eagle’s Medium (DMEM) with 10% FBS (Gibco, Dreieich, Germany) under standard cell culture conditions (37 °C and 5% CO_2_). The RSL3 (Sigma, Munich, Germany) was dissolved in dimethyl sulfoxide (DMSO) at a 10 mmol/L stock concentration and stored at −20 °C.

### 2.2. Cell Viability Assay

The LN229 cells were seeded in 96-well plates (5000 cells/well). Treatments were performed at different concentrations for 24 h, and the controls received 0.5% DMSO only. The proportion of viable cells was determined by 3-(4,5-dimethylthiazol-2-yl)-2,5-diphenyltetrazolium bromide (MTT, Sigma, Munich, Germany) assays following the manufacturer’s recommendations. Absorbance was examined by an Infinite F200 pro Microplate Absorbance Reader (Tecan, Maennedorf, Switzerland).

### 2.3. Cell Migration Assay

Three thousand cells were cultured without serum for 24 h and then seeded onto a transwell chamber, which was subsequently placed in a 24-well plate for an additional 24 h. Following the incubation period, the cells on the bottom side of the chamber were stained with crystal violet to visualize and quantify cell attachment and migration. A total of five photographs were taken for each transwell chamber, capturing different areas of the cells on the bottom side. The images were then used to count and analyze the number of cells in each photograph. This allowed for a comprehensive assessment of the cells’ behavior and distribution under the specified experimental conditions.

### 2.4. Lipid Peroxidation Assay

Cells were seeded on the cover slides and treated 24 h before performing the lipid peroxidation assays. The cells were stained with BODIPY^®^ 581/591 C11-Reagens (Thermo Fisher, Bremen, Germany) for 30 min and washed with PBS. The cells were either fixed by 30% PFA to perform fluorescence imaging or trypsinized into single cells to perform fluorescence-activated cell sorting (FACS). The FACS parameters were set according to the BODIPY^®^ 581/591 C11-Reagens kit’s guidance, and each group had 20,000 cells that went through the flow cytometer. The compensation was set as follows: FL1 −0.8% FL2; FL2 −29.8% FL1; FL2 −0.0% FL3; FL3 −19.3% FL2; FL3 −0.0% FL4; and FL4 −16.4% FL3. The FACS fluorescence images were taken using a Zeiss LSM780 fluorescence microscope (Zeiss, Munich, Germany) and analyzed using Image J software version 1.53k (National Institutes of Health, Bethesda, MD, USA). The FACS results were analyzed by FlowJo version 10.4 (FlowJo Inc., Ashland, OH, USA).

### 2.5. Oil Red O Staining Assay

The Oil Red O stain (1320-06-5, Sigma, Munich, Germany) was pre-prepared 10 min before the experiments. Cells were seeded into a 6-well plate for 24 h and received Oil Red staining for 1 min, and they were subsequently washed with 60% isopropanol for 15 s. Images were taken with a Nikon microscope (Melville, NY, USA).

### 2.6. RNA Isolation and Rt-PCR Assay

The RNA samples were isolated using Trizol Reagent (15596026, Thermofisher, Waltham, MA, USA) and chloroform and subsequently purified by isopropanol and cleaned with 70% ethanol. The purified RNA was eluted in nuclease-free water. The RT-PCR assay utilized the following materials: the total RNA extracted from the LN229 parental and TMZ-resistant cell lines, and the reverse transcription was completed using a Takara reverse transcription kit (RR037B, Takara, Saint-Germain-en-Laye, France). The ALDH1A3 forward primer (CACCTTCCACGGCCCCGTTAGCGG) and reverse primer (AAACCCGCTAACGGGGCCGTGGAA) were prepared for target amplification. A real-time PCR master mix (RR066A, Takara, Saint-Germain-en-Laye, France) was used and it contained all the necessary components for the PCR amplification. B-actin (Fw: GAGCTACGAGCTGCCTGACG, Rev: GTAGTTTCGTGGATGCCACAGGAC) was used as the control. The instrument used for amplification and data analysis was a Zytomed.

### 2.7. Western Blotting

The cells were lysed in RIPA buffer with phosphatase inhibitors (5 mM sodium orthovanadate), and the protein lysates were separated using 10% SDS-PAGE and transferred onto a PVDF membrane. The membrane was blocked in 5% BSF and subsequently incubated with primary antibodies at 4 °C overnight. After washing three times, each for 5 min, in Tris-buffered saline containing 1% Tween-20 (TBST), the membrane was then incubated with peroxidase-conjugated goat anti-rabbit IgG (7074P2, CST, Leiden, Germany) and visualized using a super ECL detection reagent (GERPN2106, sigma, Munich, Germany). The anti-ALDH1A3 (1:2000, Rabbit polyclonal,) was obtained from Abcam (ab129815, Berlin, Germany), and the anti-p-AKT (#4060, 1:3000, Rabbit monoclonal) and anti-AKT (5112S, 1:3000, Rabbit) were obtained from Cell Signaling Technology (Leiden, Germany). Vinculin (abcam, ab129002) was used as the loading control. Quantification of the results was performed using ImageJ version 1.53k.

### 2.8. Colony Formation Assay

The LN229 glioblastoma cells were seeded in 10 cm dishes at a low density (500 cells per well) and incubated for 14 days. After the incubation period, the cells were washed with phosphate-buffered saline (PBS) and fixed with 4% paraformaldehyde for 15 min at room temperature. Following fixation, the cells were stained with crystal violet solution (0.5% crystal violet in 20% methanol) for 10 min. The excess stain was washed off with distilled water, and the plates were air-dried. Photos were taken with a smartphone (Lens: 26 mm; Pixel Pitch: 1.7 µm).

### 2.9. Cell Viability Assays

The LN229 cells were seeded in 96-well plates (5000 cells/well). Treatments were performed at different concentrations for 24 h, and the controls received 0.5% DMSO only. The proportion of viable cells was determined using a 3-(4,5-dimethylthiazol-2-yl)-2,5-diphenyltetrazolium bromide (MTT, Sigma, Munich, Germany) assay following the manufacturer’s recommendations. Absorbance was examined by an Infinite F200 pro Microplate Absorbance Reader (Tecan, Maennedorf, Switzerland).

### 2.10. Statistical Analysis

Three independent experiments for each assay were conducted to validate the results. T-tests were used for the normally distributed data from the two unpaired groups. GraphPad Prism 8 (GraphPad Software Inc.; San Diego, CA, USA) was used to perform the analysis, and *p*-values of <0.05 were regarded as statistically significant.

## 3. Results

### 3.1. TMZ Long-Term Treated LN229 Cells

Ten clones of the LN229 cells were treated with gradually increasing concentrations of TMZ. Briefly, the cells were treated with TMZ for five days, followed by three weeks in a standard medium. The TMZ concentration was raised from 100 μM to 500 μM per cycle (Figure 1A). Five cell lines (TMZ-R#1, #2, #4, #7, and #8) survived the procedure and developed resistance to TMZ treatment (Figure 1B), and they were used for further experiments. The long-term treatment with TMZ resulted in morphological changes, with a transition from spindle-shaped to oval-shaped cells with cytopodia (Figure 1C). Additionally, the TMZ-resistant cells exhibited the formation of transparent vacuole structures.

Further investigation of their functional characteristics revealed that the TMZ long-term-treated cells exhibited enhanced migratory activity (Figure 2A,B). However, the colony formation assays and cell viability tests conducted over a period of 96 h demonstrated that their proliferation decreased compared to the parental LN229 cells (Figure 2C,D).

### 3.2. The TMZ Long-Term Treated Cells Formed Higher Amounts of Lipid Droplets and Showed Elevated Lipid Peroxidation but Did Not Respond to RSL3-Induced Ferroptosis

Previously, we observed that TMZ treatment led to the accumulation of reactive oxygen species (ROS), indicating a potential therapeutic target for triggering ferroptosis. Building on this finding, we investigated lipid metabolism and lipid peroxidation in TMZ-resistant cell lines in further detail. Oil Red O staining of the TMZ long-term treated cells revealed a significant increase in the accumulation of lipid droplets as the cause for the cytoplasmic vacuoles in these cells (Figure 3A). To further investigate the lipid peroxidation (LPO) state in the TMZ long-term treated cells, we stained the cells with BODIPY 581/591 dye. The BODIPY staining results indicated that long-term TMZ treatment induced a higher level of LPO compared to the parental LN229 cells (Figure 3B,C). When we treated the cells with the ferroptosis inducer RSL3, the parental LN229 cells showed a pronounced ferroptotic response, whereas all the TMZ long-term treated cells did not show ferroptosis (Figure 3D). Using Western blotting analysis, we next verified the ALDH1a3 in the TMZ long-term treated cells and found a down-regulation of the protein compared with the parental LN229 cells (Figure 3E).

### 3.3. TMZ Withdrawal Recovered ALDH1A3 Expression and Sensitivity to RSL3

After removal of the TMZ, the presence of the lipid droplets remained stable in all the TMZ long-term treated cells (Figure 4A). However, using Western blotting analysis, two out of the five clones (#4 and #8) showed re-expression of the ALDH1A3 protein (Figure 4B) that had been transcriptionally upregulated since the TMZ withdrawal (Figure 4C). Only these two clones also exhibited a reversal of the sensitivity to RSL3 (Figure 4D).

### 3.4. ALDH1A3 and Sensitivity to RSL3 Are Regulated by EGFR-Dependent Akt-Activation

We observed a down-regulation of Akt-phosphorylation in the TMZ long-term treated cells (Figure 5A). After TMZ withdrawal, Akt was phosphorylated, but only in two of the clones (#4 and #8) (Figure 5B) that exhibited ALDH1A3 re-expression and RSL3 sensitivity after TMZ removal. To learn more about the upstream regulation of ALDH1A3, the parental LN229 cells as well as the #4 and #8 cells were treated with and without AG1478, a specific EGFR tyrosine kinase inhibitor. With the AG1478 treatment, both Akt phosphorylation and ALDH1A3 expression were inhibited in the LN229 cells, as well as the #4 and #8 cells (Figure 5C), and ALDH1A3 was regulated at the transcriptional level (Figure 5D).

## 4. Discussion

Glioblastoma (GBM) is one of the most lethal brain tumors, with limited treatment approaches. Temozolomide (TMZ) is commonly used in the standard therapy for GBM, but its efficacy is often transient due to the development of resistance. A majority of patients experience recurrent GBM growth within two years [1]. Consequently, there is an urgent need to explore new treatment strategies to overcome therapy resistance.

Previously, we demonstrated that ALDH1a3 enhanced TMZ resistance in GBM and that this process was dependent on autophagy [10]. At the same time, we showed that ALDH1A3 sensitizes GBM cells to autophagy-dependent ferroptosis [19]. We could also show that ALDH1A3 and the ferroptosis elements were upregulated in relapsed GBMs [11]. By targeting the ALDH1A3-mediated autophagy-dependent ferroptosis pathway, we might be able to develop innovative and effective treatments to combat the therapy resistance in this devastating disease.

The aim of the present study was to investigate the potential of using the ferroptosis inducer RSL3 to generate ferroptotic cell death in GBM cells that were resistant to TMZ. Therefore, we generated several LN229 cell clones that were resistant to TMZ. Surprisingly, these TMZ-resistant LN229 clones also showed resistance to ferroptosis induction even though lipid peroxidation was evident. Upon further investigation, we identified the downregulation of the ALDH1a3 gene in the TMZ-resistant LN229 clones. Remarkably, when ALDH1A3 was up-regulated, these clones became re-sensitized to ferroptosis as induced by RSL3. Notably, this phenomenon appeared to be regulated by the EGFR-dependent PI3K pathway since the phosphorylation of Akt was EGFR-dependent and up-regulated specifically in the ALDH1a3-high clones.

Our previous results showed that ALDH1A3 is highly expressed in GBM cells, leading to resistance against TMZ treatment, but it is degraded by TMZ-induced autophagy [10]. Additionally, we observed that TMZ treatment induced the accumulation of reactive oxygen species (ROS) in GBM cells [8]. However, the expression of ALDH1A3 sensitized the GBM cells to ferroptosis, and this process required autophagy [19]. Therefore, the expression of ALDH1A3 appeared to be a key target in the crosstalk between TMZ-treatment and the induction ferroptosis in GBM. Here, we further proved that ALDH1A3 played an important role in therapy resistance and presented a new therapy approach to GBM. We showed that TMZ-resistant clones with low ALDH1A3 expression were resistant to RSL3, despite the presence of LDs and ROS accumulation. Furthermore, we could also demonstrate that when the expression of ALDH1A3 was reversed, the LDs accumulated TMZ-resistant clones and became sensitive to RSL3. These findings revealed the possibility of inducing ferroptosis in relapsed GBMs.

In GBM, the presence of lipid droplets (LDs) has been found to be closely linked to prognosis and treatment outcomes. Feng Gend et al. [20] demonstrated that LDs are highly accumulated in GBM cells but not in normal brain tissues or low-grade gliomas. GBM cells appear to utilize LDs to store excess fatty acids and cholesterol, thereby protecting themselves from lipotoxicity and endoplasmic reticulum (ER) stress. Furthermore, it has been shown that oxidative stress can induce the accumulation of LDs [21]. These results indicated that the TMZ treatment induced LD formation in GBM, and it potentially could be a therapy target for the ferroptosis inducer RSL3. However, our TMZ-resistant clones showed resistance to RSL3, which could be attributed to the significant role of LDs in cancer cells. These LDs act as scavengers for reactive oxygen species (ROS), offering protection to cancer cells against oxidative-stress-induced damage [22]. This protective mechanism enables cancer cells to survive and resist the oxidative stress induced by therapeutic treatments. These findings could indicate an alternative mechanism to explain RSL3 resistance in TMZ-resistant clones.

In our previous results, a five-day TMZ treatment suppressed the expression of ALDH1A3, followed by a rebound to higher levels after TMZ withdrawal. In this study, a similar pattern emerged, with ALDH1A3 expression being suppressed following TMZ treatment, and notably, two out of five TMZ-resistant cell lines showed a substantial recovery of ALDH1A3 expression within one month of TMZ withdrawal. These findings indicated that the recovery of ALDH1A3 could depend on the duration of TMZ withdrawal. During prolonged TMZ exposure, there is a drastic reduction in cell numbers, leading to the survival of TMZ-resistant clones through selective single-cell-like proliferation. In this case, the intrinsic heterogeneity within glioblastoma cells was a crucial consideration as it could have significantly impacted the experimental reproducibility and the interpretation of our results. Therefore, acknowledging the existence of heterogeneity is imperative when drawing conclusions from our findings.

Interestingly, we observed a correlation between the EGFR pathway activation and the recovered expression of ALDH1A3. Two ALDH1A3-recovered, TMZ-resistant clones exhibited activated Akt signaling, and the expression of ALDH1A3 could be down-regulated by EGFR-inhibition using the highly specific EGFR pathway inhibitor AG1478. The EGFR pathway promotes cell proliferation and metastasis and drives tumor recurrence. Aberrant EGFR signaling, often caused by overexpression, gene mutations, or amplification, is frequently observed in various cancer types, including GBM. The EGFR pathway exhibits abnormal activation in approximately 30% of glioblastoma patients, and it is associated with poor prognosis and tumor invasion. A study conducted by Nobuharu Inaba revealed that inhibiting the EGFR pathway led to a deceleration in cell proliferation and did not enhance sensitivity to TMZ treatment [23]. Gong et al. provided evidence that EGFR deficiency is associated with chemotherapy resistance in glioblastoma [24]. These studies support our findings, suggesting that the EGFR pathway may indeed be suppressed in TMZ-resistant cell lines. Moreover, phosphorylated AKT triggers the production of nitric oxide (NO), thus mediating cell proliferation, migration, ROS metabolism, and therapy resistance [25]. Additionally, based on our previous results, we found ALDH1A3 to be highly expressed in relapsed GBM, further supporting its role in tumor recurrence. These compelling findings collectively suggest a hidden connection between an activated EGFR pathway and ALDH1A3 expression in relapsed GBM, making them potential targets for ferroptosis. However, due to our limited findings, confirming EGFR’s regulation of ALDH1A3 at the transcriptional level remains challenging. Additional experiments are necessary to unravel the underlying mechanism linking an activated EGFR pathway and the regulation of ALDH1A3. Targeting these pathways could offer a novel therapeutic strategy to combat treatment-resistant and recurrent GBM. Further research and exploration of these pathways might lead to the development of innovative therapies aimed at improving outcomes for patients with recurrent GBM.

In conclusion, our study sheds light on the complex interplay between lipid metabolism, ALDH1A3 expression, and TMZ resistance in glioblastoma. Nevertheless, our findings were constrained by the in vitro experiments conducted in an established cell line and the limited insights. Further investigations are essential for exploring the hidden mechanisms related to manipulating the EGFR pathway, therapy resistance, and the induction of ferroptosis in glioblastoma. Understanding these mechanisms may lead to the development of novel therapeutic strategies to overcome TMZ resistance and improve treatment outcomes in relapsed glioblastoma patients.

## Figures and Tables

**Figure 1 cells-12-02522-f001:**
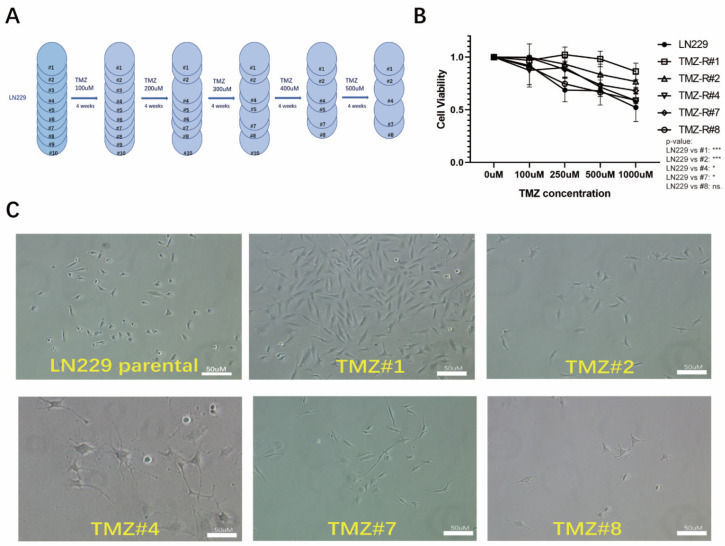
(**A**) The construction of the TMZ-resistant LN229 clones. (**B**) The cell viabilities of the LN229 cells and the five LN229 TMZ-resistant clones under increasing dosages of TMZ (100 uM, 250 uM, 500 uM, and 1000 uM) (* *p* < 0.05 and *** *p* < 0.001). (**C**) The cell morphologies of the LN229 cells and the five LN229 TMZ-resistant clones after 6 months of TMZ treatment.

**Figure 2 cells-12-02522-f002:**
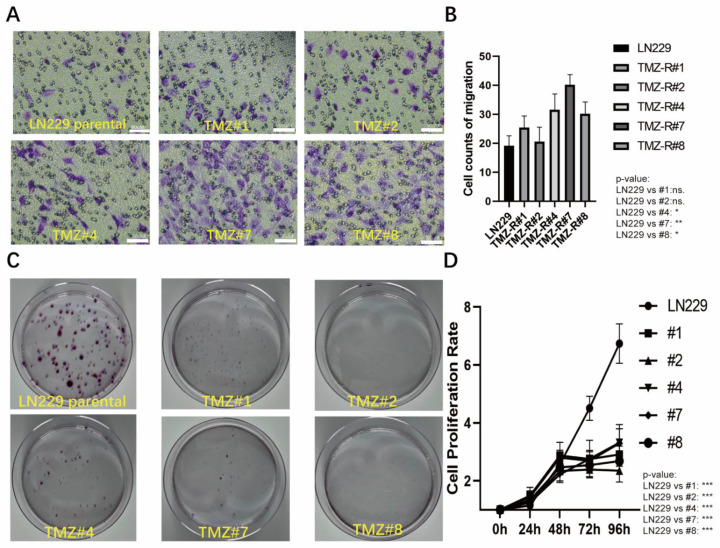
(**A**) The LN229 cells and the five LN229 TMZ-resistant clones were seeded into a Transwell chamber for the migration assay, and the cells on the bottom of chamber that were stained by the crystal violet were considered to be migrated cells. (**B**) Quantification of A (* *p* < 0.05, ** *p* < 0.01, and *** *p* < 0.001). (**C**) Clone formation assay of the LN229 cells and the five LN229 TMZ-resistant clones under a normal medium (without TMZ). (**D**) Cell proliferation rates of the LN229 cells and the five clones (TMZ #1, TMZ #2, TMZ #4, TMZ #7, and TMZ #8) after 24 h, 48 h, 72 h, and 96 h, as measured by the MTT assay (* *p* < 0.05, ** *p* < 0.01, and *** *p* < 0.001).

**Figure 3 cells-12-02522-f003:**
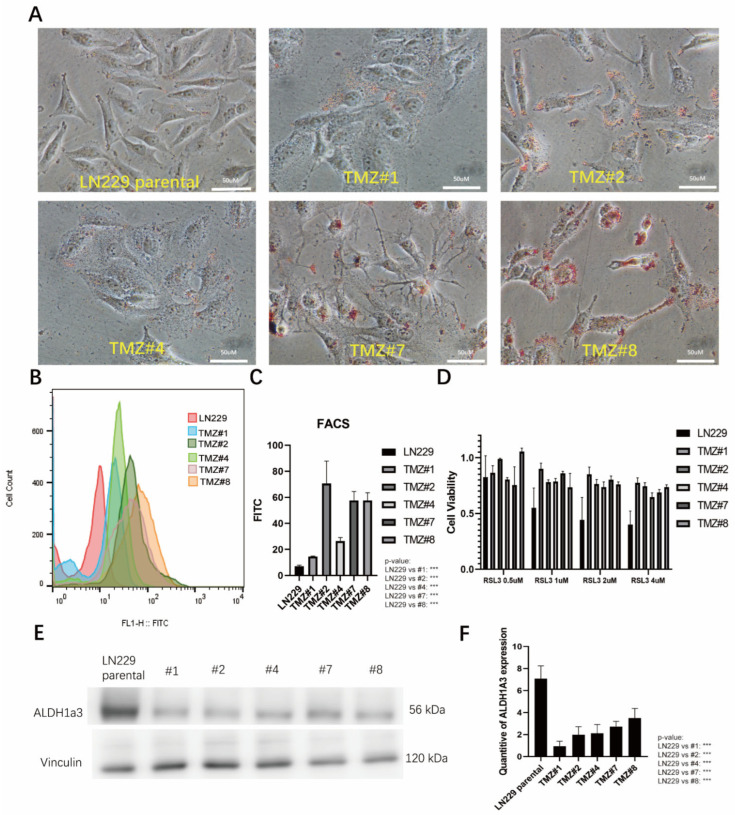
(**A**) The LN229 cells and the five LN229 TMZ-resistant clones were stained by Oil red O under a normal medium (TMZ removed before the cell seeding). After Oil Red O staining, the images showed bright red coloration in the lipid droplets. The lipid droplets were uniformly distributed and exhibited well-defined round shapes. (**B**) The BODIPY staining in the LN229 cells and the five LN229 TMZ-resistant clones as analyzed by FACS. X axis, density of fluorescence; Y axis, cell count of the fluorescent cells. (**C**) Quantification of B. (**D**) The cell viabilities of the LN229 cells and the five LN229 TMZ-resistant clones under the RSL3 treatments (0.5 uM, 1 uM, 2 uM, and 4 uM) (*p*-value: LN229 vs. TMZ #1, #2, #4, #7, and #8 in 4 uM: (*** *p* < 0.001)). (**E**) The protein expression of ALDH1a3 in the LN229 cells and the five LN229 TMZ-resistant clones as analyzed by Western blotting. (**F**) Quantification of E, measured and analyzed using ImageJ version 1.53k. *** *p* < 0.001.

**Figure 4 cells-12-02522-f004:**
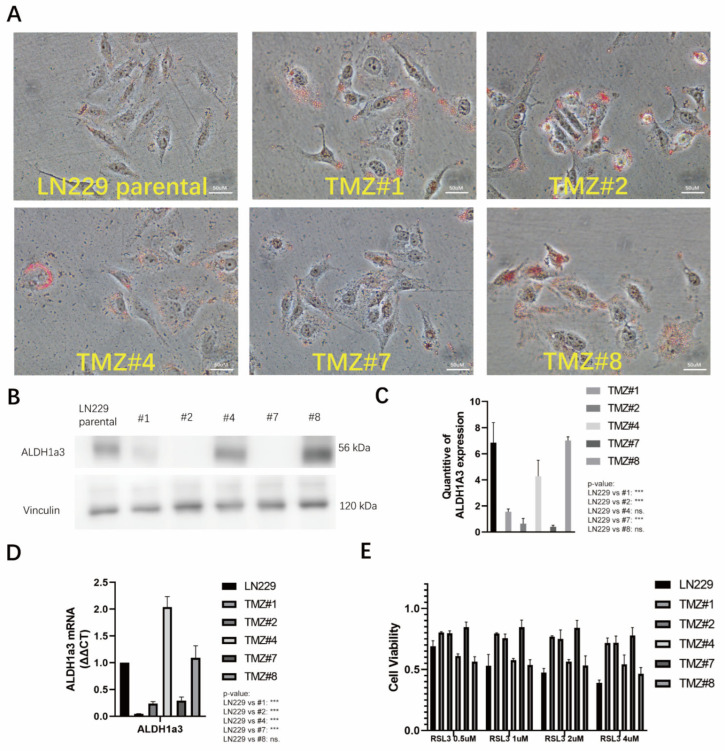
(**A**) After one-month of TMZ withdrawal, the LN229 cells and the five LN229 TMZ-resistant clones were stained with Oil red O. After Oil Red O staining, the images showed bright red coloration in the lipid droplets. (**B**) Protein expression of ALDH1a3 in the LN229 cells and the five LN229 TMZ-resistant clones after TMZ withdrawal, as analyzed by Western blotting. (**C**) Quantification of B, measured and analyzed using ImageJ version 1.53k (*** *p* < 0.001). (**D**) Quantification of the ALDH1a3 mRNA in the LN229 cells and the five LN229 TMZ-resistant clones after TMZ withdrawal, as analyzed by RT-PCR (*p*-value: LN229 vs. TMZ #1, #2, and #7: ***; LN229 VS TMZ #4: *** (*** *p* < 0.001)). (**E**) Cell viabilities of the LN229 cells and the five LN229 TMZ-resistant clones under the RSL3 treatments after TMZ withdrawal (0.5 uM, 1 uM, 2 uM, and 4 uM) (*p*-value: LN229 vs. TMZ #1, #2, and #7: *** (*** *p* < 0.001)).

**Figure 5 cells-12-02522-f005:**
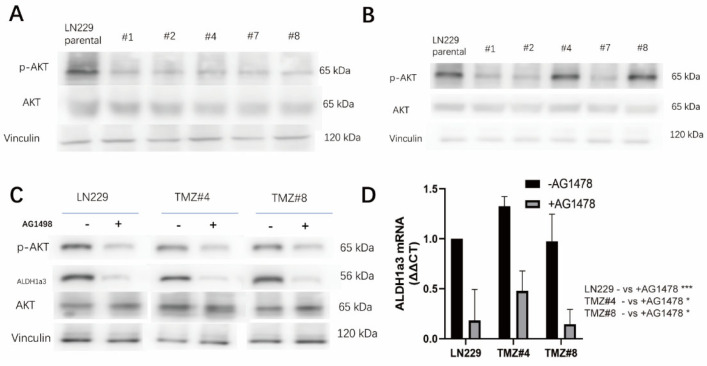
(**A**) The phosphorylation state of AKT in the LN229 cells and the five LN229 TMZ-resistant clones under the TMZ culture medium, as analyzed by Western blotting. (**B**) The phosphorylation state of AKT in the LN229 cells and the five LN229 TMZ-resistant clones after TMZ withdrawal, as analyzed by Western blotting. (**C**) The phosphorylation state of AKT and the protein expression of ALDH1a3 in the LN229 cells and the five LN229 TMZ-resistant clones after TMZ withdrawal, with/without AG1498 treatment, as analyzed by Western blotting. (**D**) Quantification of the ALDH1a3 mRNA in the LN229 cells and in the TMZ #4 and TMZ #8 cells with/without AG1478, as analyzed by RT-PCR (*p*-value: LN229: *; TMZ #4: *; and TMZ #8: * (* *p* < 0.05 and *** *p* < 0.001)).

## Data Availability

There’s no sharing data.

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
