# Peer review of "Enhanced Sensitivity to ALDH1A3-Dependent Ferroptosis in TMZ-Resistant Glioblastoma Cells"

_cells, 2023, doi:10.3390/cells12212522_

Round 1

Reviewer 1 Report

Comments and Suggestions for Authors The subject of this article is my favorite, and the researchers did a good job on the project, but for the final acceptance, it is necessary to do the following: 1- The article should be reviewed in terms of grammar and writing. 2- In the introduction section, the necessity of doing the work should be explained better. 2- The discussion part should be explained more.  

Comments on the Quality of English Language

The article should be checked for grammar and writing.

Author Response

We would like to express our gratitude for the time and effort you invested in evaluating our manuscript. The comments have been extremely insightful and beneficial. In response to the Reviewer's comments, we have made a number of significant revisions. Specifically, we have corrected numerous grammatical errors (1) and clarified sections that were previously misunderstood. In addition, we have included the manuscript's specific focus in the introduction (2) and provided expanded explanations for specific details in the discussion (3).

Reviewer 2 Report

Comments and Suggestions for Authors

Attention should be paid to the role of nitric oxide in the regulation of Akt phosphorylation and blocking of the EGFR signaling pathway in further studies of glioblastoma.

Author Response

We are very grateful to the Reviewer's comment. We revised our discussion section to incorporate the significant role of nitric oxide in EGFR signaling. We will continue to explore and focus on the role of nitric oxide in our future studies.

Reviewer 3 Report

Comments and Suggestions for Authors

1Analysis of ALDH1A3 gene expression in TMZ-resistant LN229 cell lines was performed in the manuscript. The manuscript needs to be carefully revised from Abstract to Reference. While the manuscript is suitable as a topic, further analyses on ferroptosis and PI3K/AKT pathways need to be performed. However, some areas that need to be corrected in the publication are listed below;

11-      For the manuscript to be more understandable, it is necessary to make corrections to the language.

22-      The manuscript's expression "TMZ resistant" should be corrected to "TMZ-resistant".

33-      In line 48, indicate whether high or low expression levels of ALDH1A3 promote the maintenance of glioma stem-like cells.

44-      The mechanism of action of the PI3K/AKT pathway on the expression level of the ALDH1A3 needs to be further clarified in the introduction section of the manuscript.

55-      In line 105, is it possible to isolate RNA with just chloroform? Isopropanol is used for the precipitation of nucleic acids.

66-      In line 110, RT-qPCR analysis, the sequences of the forward and reverse primers specified for the determination of the RNA expression level of the ALDH1A3 gene do not match at all with the sequence data found in "Ensembl genome browser 110" for the ALDH1A3 gene sequence. However, excluding the reverse primer, the sequence of the B-actin Forward primer with the sequence 5'-CGCGAGAAGATGACCCAGATC-3' is mismatched again. The 5' and 3' aspects of the indicated sequences are not specified. In this case, it is a question of how the authors were able to perform RT-qPCR analyses with these primers.

77-      The material and method section should open a separate heading for Western Blot. However, Vinculin expression analysis was performed in the results section. On the other hand, why Vinculin was chosen here is not specified in the introduction. Also, the materials and methods section did not mention which Vinculin antibody was used for expression analysis.

18-      The image in Figure 3B was probably taken with a camera, which is enlarged from beginning to end.

Comments on the Quality of English Language

Moderate editing of English language required.

Author Response

Unquestionably, the Reviewer's insights have improved the quality of the manuscript, and we greatly value his/her expertise and considerate suggestions. The following are the point-by-point responses:

1-      For the manuscript to be more understandable, it is necessary to make corrections to the language.

We have rectified numerous grammar errors and clarified previously misunderstood sections of the writing (see also comments to Reviewer 1).

2-      The manuscript's expression "TMZ resistant" should be corrected to "TMZ-resistant".

We have made corrections.

3-      In line 48, indicate whether high or low expression levels of ALDH1A3 promote the maintenance of glioma stem-like cells.

High expression of ALDH1A3 promote the maintenance of stem cell state, we’ve corrected it.

4-      The mechanism of action of the PI3K/AKT pathway on the expression level of the ALDH1A3 needs to be further clarified in the introduction section of the manuscript.

We have incorporated in the introduction various regulatory mechanisms governing the expression of ALDH1A3. Additionally, we have highlighted the need for further studies to confirm the correlation between the EGFR pathway and ALDH1A3 expression in the discussion section.

5-      In line 105, is it possible to isolate RNA with just chloroform? Isopropanol is used for the precipitation of nucleic acids.

There’s a missing information, now it’s corrected.

6-      In line 110, RT-qPCR analysis, the sequences of the forward and reverse primers specified for the determination of the RNA expression level of the ALDH1A3 gene do not match at all with the sequence data found in "Ensembl genome browser 110" for the ALDH1A3 gene sequence. However, excluding the reverse primer, the sequence of the B-actin Forward primer with the sequence 5'-CGCGAGAAGATGACCCAGATC-3' is mismatched again. The 5' and 3' aspects of the indicated sequences are not specified. In this case, it is a question of how the authors were able to perform RT-qPCR analyses with these primers.

We sincerely apologize for the oversight, which occurred due to an inadvertent copy-paste error. We have corrected the mistake.

7-      The material and method section should open a separate heading for Western Blot. However, Vinculin expression analysis was performed in the results section. On the other hand, why Vinculin was chosen here is not specified in the introduction. Also, the materials and methods section did not mention which Vinculin antibody was used for expression analysis.

We appreciate the Reviewer highlighting our negligence. We have added Western Blotting in Material and Methods and mentioned vinculin as the loading control.

8-      The image in Figure 3B was probably taken with a camera, which is enlarged from beginning to end.

We have adjusted the graphics of figure 3B.

Reviewer 4 Report

Comments and Suggestions for Authors

The manuscript  entitled “Enhanced Sensitivity to ALDH1A3-dependent Ferroptosis in TMZ-Resistant Glioblastoma Cells” is an interesting study by Wu et al., In their investigation, Wu et al. delve into the intricate role of the ALDH1A3 gene, which was shown to have a role in TMZ resistance, and its significance in ferroptosis. Interestingly, the GBM cells that resisted TMZ also exhibited an aversion to ferroptosis. This resistance could be reversed when ALDH1A3 expression was reinstated, suggesting a critical interplay between ALDH1A3 and ferroptosis. The modulation of ALDH1A3 seems to be under the influence of the EGFR-dependent PI3K pathway. Additionally, lipid droplets, known for their protective attributes against oxidative stress, may be intertwined in this resistance mechanism. Although this is an important finding, the authors have to address the following concerns below:

The authors have used only one commercially available GBM line LN229. What is the MGMT status of this line? Any knowledge about mutational status or subtype known? 

Figure 1A: What are the levels of ALDH1A3 at varying time points during the development of TMZ resistance? Western blots could be done to show expression at each timepoint. 

Figure 1C: The cells are not visible. Very small images, no scale bars, contrast should be increased. 

Figure 2B and D: The authors should indicate the appropriate p values 

Figure 2C: How was this study done? Magnification of images should be shown. More robust method could be used, perhaps Ki-67 staining. Method for colony formation also not in methods.

Figure 2D: Better graph should be shown, poorly stratifies data. Need statistics between groups and data needs to be clearly displayed. 

Figure 3: Clear rationale to link to findings in first two figures should be mentioned. Rationale must be developed for why lipid peroxidation was explored. 

Figure 3A: Scale bars in each image? 

Figure 3B: FACS not in methods? Explain what is done? What is normalization for flow? The axes labels are missing. Perhaps histogram plots comparing MFI shift would be better. 

Figure 3D: Need to better stratify data and show comparisons. 

Figure 3E: Need to quantify the western blots, differences are not visually clear due to variable loading control. 

Figure 4A: The authors employ Oil Red O staining for LN229 and LN229 TMZ-resistant clones but fail to elaborate on its significance. This omission, in both the main content and figure caption, hinders effective interpretation and comprehension of the paper.

Figure 4B: The authors note that TMZ withdrawal can induce ALDH1A3 expression and RSL3 sensitivity recovery. However, only two out of five clones demonstrated ALDH1A3 re-expression post-TMZ withdrawal. This observation implies that such an effect might not be consistent across all GBM clones, potentially being specific to particular cellular clones or populations. Thus, the evidence seems insufficient to assert that TMZ withdrawal universally restores ALDH1A3 in recurrent GBMs. There is no quantification of the western blot expression of ALDH1A3 for the respective clones, which limits the objective assessment of protein expression levels. 

Figure 5B: The focus on only two clones (#4 and #8) that regain Akt phosphorylation after TMZ withdrawal raises questions about the universality of this observed phenomenon. Does this imply that only certain cellular clones possess this capability, or are there external factors not yet accounted for?

While AG1478's impact on both Akt phosphorylation and ALDH1A3 expression is evident, one must question the specificity of this inhibitor in the given cellular context. Could AG1478 be affecting other pathways or molecules that were not investigated in this study?

Figure 5 D: Lastly, the claim that "ALDH1A3 is regulated at the transcriptional level" seems premature. A more detailed discussion or additional evidence detailing the mechanism and influencing factors in this transcriptional regulation would enhance this claim.

While the study sheds valuable light on resistance pathways, a pressing concern arises from the observed heterogeneity in GBM cells. Notably, only a fraction of cells reverted to being sensitive to RSL3 following TMZ cessation, casting doubts over the broad applicability of this therapeutic angle for GBM. This should be discussed in detail. 

Typo in abstract: "Recover of ALDH1A3 expression" should be recovery. 

Typo in Figure 3 caption: "Westernblotting" as one word – should be separate. 

Author Response

We are extremely grateful for the reviewer's insightful comments and observations. The following are the point-by-point responses:

The authors have used only one commercially available GBM line LN229. What is the MGMT status of this line? Any knowledge about mutational status or subtype known?

LN229 harbor MGMT deficiency and p53 mutation, which we have added in the Materials and Methods part.

Figure 1A: What are the levels of ALDH1A3 at varying time points during the development of TMZ resistance? Western blots could be done to show expression at each timepoint.

This is a good query; we have examined the expression of ALDH1A3 before and after TMZ withdrawal, as shown in figure 3E and 4B. We also examined the expression of ALDH1A3 and discovered that another TMZ-res cell line also showed a recovery of ALDH1A3, however, we decided not to include these results from another cell line in the manuscript.

Figure 1C: The cells are not visible. Very small images, no scale bars, contrast should be increased.

We have resolved this issue.

Figure 2B and D: The authors should indicate the appropriate p values

We are very thankful for the Reviewer's reminder. Before, we had the p-value in the figure legend, but now for better display we have added p-value in the figures.

Figure 2C: How was this study done? Magnification of images should be shown. More robust method could be used, perhaps Ki-67 staining. Method for colony formation also not in methods.

We appreciate the Reviewer pointing out our carelessness. We have added the method of colony formation in Material and Methods. In figure 2D, we have cell viability results done by MTT assay that has now also been added to Material and Methods.

Figure 2D: Better graph should be shown, poorly stratifies data. Need statistics between groups and data needs to be clearly displayed.

We have enlarged the figure and added the p-values.

Figure 3: Clear rationale to link to findings in first two figures should be mentioned. Rationale must be developed for why lipid peroxidation was explored.

We have added a comment at the beginning of Results 2.

Figure 3A: Scale bars in each image?

We have added where missing.

Figure 3B: FACS not in methods? Explain what is done? What is normalization for flow? The axes labels are missing. Perhaps histogram plots comparing MFI shift would be better.

The details of FACS experiments have been added to Material and Methods (2.4). The FACS results have now been converted into histograms.

Figure 3D: Need to better stratify data and show comparisons.

We have adjusted and added p-values in the figure.

Figure 3E: Need to quantify the western blots, differences are not visually clear due to variable loading control.

The quantified results of western blots are now shown in figure 3F and Figure 4C.

Figure 4A: The authors employ Oil Red O staining for LN229 and LN229 TMZ-resistant clones but fail to elaborate on its significance. This omission, in both the main content and figure caption, hinders effective interpretation and comprehension of the paper.

We have now added more information to the figure legends.

Figure 4B: The authors note that TMZ withdrawal can induce ALDH1A3 expression and RSL3 sensitivity recovery. However, only two out of five clones demonstrated ALDH1A3 re-expression post-TMZ withdrawal. This observation implies that such an effect might not be consistent across all GBM clones, potentially being specific to particular cellular clones or populations. Thus, the evidence seems insufficient to assert that TMZ withdrawal universally restores ALDH1A3 in recurrent GBMs. There is no quantification of the western blot expression of ALDH1A3 for the respective clones, which limits the objective assessment of protein expression levels.

In our previous findings, a 5-day TMZ treatment led to a suppression of ALDH1A3 expression, followed by a rebound to higher levels upon TMZ withdrawal. In the current study, a similar trend was observed, Notably, within one month of TMZ withdrawal, two out of five TMZ-resistant cell lines exhibited a significant recovery in ALDH1A3 expression. With longer time duration of TMZ withdrawal, the expression of ALDH1A3 in another cell line (TMZ#1) recovered slightly (data not shown in the manuscript). These observations hint at a potential dependence of ALDH1A3 recovery on the duration of TMZ withdrawal.

Another explanation is, during prolonged TMZ exposure, there is a significant reduction in cell numbers, leading to the survival of TMZ-resistant clones through selective single-cell-like proliferation. This phenomenon underscores the intrinsic functional heterogeneity even within pseudo-clonal LN229 glioblastoma cells, a crucial factor that can have a profound effect on experimental interpretation of results. On the other hand, experimental investigation of the specific tumor biology leading to this functional heterogeneity could provide new insights into the molecular basis of therapy resistance and open the door to novel therapeutic approaches.

Figure 5B: The focus on only two clones (#4 and #8) that regain Akt phosphorylation after TMZ withdrawal raises questions about the universality of this observed phenomenon. Does this imply that only certain cellular clones possess this capability, or are there external factors not yet accounted for?

We are extremely thankful for this important question regarding our study. We acknowledge the limitation in our current study, where we concentrated on these two clones to explore the recovery of Akt phosphorylation after TMZ withdrawal, and have mentioned this in the discussion. This was an initial investigation aiming to establish a foundation for understanding the underlying mechanisms. We recognize the need for a more comprehensive analysis across a broader spectrum of GBM cell lines and clones to get a better insight into the functional biology of this phenomenon. Further experiments are planned to investigate additional clones and elucidate whether this capability is unique to specific cellular clones or influenced by external factors not yet considered.

While AG1478's impact on both Akt phosphorylation and ALDH1A3 expression is evident, one must question the specificity of this inhibitor in the given cellular context. Could AG1478 be affecting other pathways or molecules that were not investigated in this study?

AG1478 is a highly selective EGFR Inhibitor.

Figure 5 D: Lastly, the claim that "ALDH1A3 is regulated at the transcriptional level" seems premature. A more detailed discussion or additional evidence detailing the mechanism and influencing factors in this transcriptional regulation would enhance this claim.

The Reviewer's comments highlight a crucial aspect that demands in-depth investigation. In response, we have elaborated our analysis of the regulation of the EGFR pathway and ALDH1A3 expression. While our results do not wholly confirm the direct transcriptional regulation of ALDH1A3 expression by the EGFR pathway, they do reveal a significant correlation between the two.

While the study sheds valuable light on resistance pathways, a pressing concern arises from the observed heterogeneity in GBM cells. Notably, only a fraction of cells reverted to being sensitive to RSL3 following TMZ cessation, casting doubts over the broad applicability of this therapeutic angle for GBM. This should be discussed in detail.

We once again thank the Reviewer for his/her observations and thoughtful suggestions. We have already discussed this comment in our response to Fig. 4B and have incorporated this nuanced perspective into the discussion.

Typo in abstract: "Recover of ALDH1A3 expression" should be recovery.

Corrected

Typo in Figure 3 caption: "Westernblotting" as one word – should be separate.

Corrected

Round 2

Reviewer 3 Report

Comments and Suggestions for Authors

With the changes noted by other reviewers, the manuscript has become better. However, a few things need to be corrected in the new changes. After these corrections, the related manuscript can be accepted.

1. The word goat, repeated in line 86, should be removed from the sentence.

2. The definition of "Rt-PCR" in line 120 should be corrected to RT-PCR. The definition of "rt-PCR" in lines 228 and 246 should be corrected to RT-PCR. Other misspellings of the definition should be corrected in the manuscript.

3. Antibodies in Section 2.1 should be moved to the newly opened Section 2.7.

4. It should be stated at what value the asterisks indicating the p-value specified in the description section of the figures indicate significance. For example; * p=0.05, ** p=0.01, **** p<0.0001 etc. Additionally, it would be more useful to indicate asterisks above the bars in the charts.

Author Response

Dear reviewer,

Thank you for pointing out the omissions in the text. We have corrected them.

Reviewer 4 Report

Comments and Suggestions for Authors

The authors have addressed the questions raised. One suggestion would be to include the p values themselves in all the figure legends. For eg., if they had a significance  p<0.01, p <0.001, then *p<0.01, **p <0.001  or whatever the value they obtained. Right now, its unclear whether *=0.05 or 0.01?

Author Response

Dear reviewer,

We appreciate your careful observation of the omissions in the text, and we have made the corrections.